# Microbubble Delivery Platform for Ultrasound-Mediated Therapy in Brain Cancers

**DOI:** 10.3390/pharmaceutics15020698

**Published:** 2023-02-19

**Authors:** Kibeom Kim, Jungmin Lee, Myoung-Hwan Park

**Affiliations:** 1Department of Chemistry and Life Science, Sahmyook University, Seoul 01795, Republic of Korea; 2Convergence Research Center, Nanobiomaterials Institute, Sahmyook University, Seoul 01795, Republic of Korea; 3Department of Convergence Science, Sahmyook University, Seoul 01795, Republic of Korea; 4N to B Co., Ltd., Seoul 01795, Republic of Korea

**Keywords:** microbubble, ultrasound, brain cancer, blood-brain barrier, drug delivery

## Abstract

The blood-brain barrier (BBB) is one of the most selective endothelial barriers that protect the brain and maintains homeostasis in neural microenvironments. This barrier restricts the passage of molecules into the brain, except for gaseous or extremely small hydrophobic molecules. Thus, the BBB hinders the delivery of drugs with large molecular weights for the treatment of brain cancers. Various methods have been used to deliver drugs to the brain by circumventing the BBB; however, they have limitations such as drug diversity and low delivery efficiency. To overcome this challenge, microbubbles (MBs)-based drug delivery systems have garnered a lot of interest in recent years. MBs are widely used as contrast agents and are recently being researched as a vehicle for delivering drugs, proteins, and gene complexes. The MBs are 1–10 μm in size and consist of a gas core and an organic shell, which cause physical changes, such as bubble expansion, contraction, vibration, and collapse, in response to ultrasound. The physical changes in the MBs and the resulting energy lead to biological changes in the BBB and cause the drug to penetrate it, thus enhancing the therapeutic effect. Particularly, this review describes a state-of-the-art strategy for fabricating MB-based delivery platforms and their use with ultrasound in brain cancer therapy.

## 1. Introduction

The blood-brain barrier (BBB) is a unique interface composed of blood capillaries in the central nervous system (CNS) that maintains homeostasis in the neural microenvironment and protects the brain parenchyma from foreign toxic substances [1,2,3,4,5,6,7]. Tight junctions between the brain capillaries and endothelial cells (ECs) prevent the transport of molecules, except for small-sized molecules and lipid molecules, into the brain. In addition to the tight junctions, the brain capillary ECs, which have extraordinarily low transcytosis rates and no fenestrations, inhibit the transcellular transport of macromolecules [8,9,10,11,12,13,14,15]. In the treatment of brain cancers, chemotherapy is a major treatment method for suppressing cancer expression to prevent cancer recurrence after surgical resection [16,17]. However, the barrier properties of the BBB and efflux transporters P-glycoproteins and breast cancer-resistant proteins that are highly expressed in BBB are the primary cause hindering the delivery of drugs to the CNS for cancer treatment [12,18]. Therefore, although several studies have been conducted on approaches for delivering anticancer drugs to the CNS, limitations such as high invasiveness, poor distribution, insufficient efficacy, and intolerable toxicity remain [12].

Due to the unique structure of the brain, the treatment of brain-related diseases has involved various technological attempts to overcome this hindrance effectively by promoting drug delivery to the CNS [2,19]. Among these attempts, thermal ablation using high temperature generated by focused ultrasound energy, whose safety and efficacy have been verified, is being actively attempted worldwide as a surgical treatment for brain diseases such as obsessive-compulsive disorder, mental disorders such as depression, and intractable pain [20,21,22,23]. Particularly, various clinical trials have been conducted to treat brain diseases effectively by promoting drug delivery to the CNS via control of the BBB through the stable and inertial cavitation phenomenon of microbubbles (MBs) that occur when focused ultrasound is applied to the cerebral blood vessels [24,25,26]. Additionally, the MB-based system is of great interest for the effective delivery of drugs across the BBB due to its non-invasive, transient, reversible, and localized properties [27,28,29,30,31,32].

In the 1930s, Dr. Karl Dussik first published a paper on the use of ultrasound to visualize cerebral ventricles in the brain by measuring reflections of ultrasound through the head and in the late 1960s, Dr. Claude Joyner first noted the development of MBs as a contrast medium, and the clinical use of MBs started after Gramiak and Shah acknowledged this subject [33,34,35]. However, the vast potential of MBs as drug-delivery vehicles was recognized only in the late 1990s [36,37,38]. The structure of the MBs comprises an inner center (core) and an outer layer (shell). MBs physically interact with the surrounding medium through stable cavitation or inertial cavitation, depending on the ultrasound intensity used. On the one hand, stable cavitation generated by ultrasound excitation causes continuous oscillation, which induces a liquid flow around the MBs. On the other hand, inertial cavitation at higher ultrasonic intensities results in the implosion or collapse of the MBs, generating violent mechanical stresses, microjets, and shock waves [17,39,40,41,42,43]. In addition, since the required acoustic energy and response differ depending on the size and concentration of microbubbles, the physical parameters of ultrasound, such as period per pulse, pulse amplitude, pulse repetition frequency, and exposure length, must be adjusted to determine the optimal ultrasound and microbubble combination [44,45]. These phenomena induce biological changes in the BBB, such as increased endocytosis/transcytosis, paracellular passage, and opening of the tight junctions, thus increasing the permeability of the BBB macrostructure for brain cancer therapy [17,46,47,48,49]. This review investigated the generation methods, constituent materials, and various studies on MBs for brain cancer treatment (Figure 1).

## 2. Structure and Composition of MBs

The structure of the MBs comprises the core and shell structures, each with different physicochemical properties. Various materials have been used as the core and shell structural components to increase the stability and efficiency of MBs for brain cancer therapy [34,50].

### 2.1. Core Structure

The first-generation MBs had low stability in solution because atmospheric air constituted the core, and they lacked a stabilizing shell. The stability was increased in the next generation of MBs by incorporating a shell; however, MBs that have an air core have low stability in the biological environment because air dissolves in the blood [38,51,52]. Therefore, in brain cancer therapy, sulfur hexafluoride (SF_6_) or perfluorocarbons, which have higher molecular weights and lower blood solubility than those of atmospheric air, are used as the core in MBs. SonoVue^®^ (SF_6_), Definity^®^ (perfluoropropane [C_3_F_8_]), and Optison^®^ (C_3_F_8_), which are commercially used, have a perfluorochemical as the material constituting the core. Additionally, other studies have utilized perfluorochemicals such as perfluorobutane (C_4_F_10_) and perfluoropentane (C_5_F_12_) as the core components [38,52]. Particularly, C_5_F_12_, a volatile gas with a boiling point of 26 °C, exists in liquid form during the MBs’ manufacturing process and forms the MBs while changing to the gaseous state at body temperature after injection [34,38,52,53].

### 2.2. Shell Structure

The shell structure comprises polymers, proteins, surfactants, or phospholipids to prevent gas leakage, breakdown, and coalescence [34,36,53,54]. Notably, the shell structure significantly reduces the surface tension of MBs, which is closely related to their stability [54,55,56]. The water and gas molecules at the MB gas–water interface form hydrogen bonds horizontally along the interface line. This creates a contraction force on the surface of the MBs, which generates surface tension in the inward direction and induces an increase in the pressure inside them. Higher pressure increases the dissolution rate of the gas, reducing the stability of the MBs. The shell structure that covers the surface of the core increases the stability of the MBs by preventing the formation of ordered hydrogen bonds and reducing the inner pressure of the MBs [28,38,53,56].

The constituent shell materials mainly used in the brain cancer therapy system are phospholipids and phospholipid derivatives. Moreover, a broad range of phospholipids with various hydrophobic chain lengths and electrostatic charges have been utilized [53,57,58]. Thus, it facilitates the customization of the system according to the characteristics of the delivered material. The hydrophobicity of phospholipids affects the drug-loading capacity and stability of the MBs [59]. Moreover, polyethylene glycol-modified phospholipid (PEG-PL) endows the MBs with a stealth effect to evade clearance by the reticuloendothelial system Phospholipids such as dipalmitoyl phosphoric acid (DPPA), dipalmitoylphosphatidylethanolamine (DPPE), dipalmitoylphosphatidylcholine (DPPC), dipalmitoylphosphatidylglycerol (DPPG), distearoylphosphatidylcholine (DSPC), and distearoylphosphatidylglycerol (DSPG) are used to form the shell of the MBs (Figure 1).

### 2.3. Multi-Functionalization of MBs

The effects of brain cancer therapy can be enhanced by modifying the MBs comprising a core and shell with cancer drugs, cancer-targeting ligands, DNA, and nanomedicine. These materials develop an association with the MBs shell via electrostatic or hydrophobic interactions, van der Waals forces, physical encapsulation, host-guest interactions, or covalent bonds [28,38,60,61]. Furthermore, functionalized MBs’ structures were designed by modifying the material properties and interactions between the MBs. Hydrophilic DNA is attached to the surface of the MBs comprising positively charged lipid shells via electrostatic interactions (Figure 2a) [62]. Lipophilic or hydrophobic drugs can be encapsulated in the phospholipid shells (Figure 2b) [63,64,65]. The host-guest interaction of avidin-biotin was used to modify the targeting ligand on the MBs’ surface (Figure 2c) [62,65]. Because the surface area or shell volume of the MBs is restricted, modification of the liposomes on the MBs’ surface provides a larger space for accommodating medicinal materials (Figure 2d) [28,66,67,68,69,70].

## 3. Fabrication Method

### 3.1. Sonication Method

This method uses high-intensity ultrasound to generate MBs after dispersing the gas constituting the core in a liquid containing an appropriate coating material that forms the shell for stabilizing the MBs. The size distribution of the generated MBs depends on the frequency, power, and pulse region of the ultrasound; however, the theoretical relationship between these parameters and fabrication protocols has not been elucidated and has been developed empirically. The material is used as the outer shell and dissolved in the aqueous solution, and sonication is applied. Gas is continuously supplied to the solution for MBs generation. After generating the MBs using sonication, centrifugation and/or filtration are required to remove the residual substances [34,71,72,73].

### 3.2. Thin-Film Hydration Method

A solution of the material used as the outer shell and dissolved in a volatile solvent is evaporated to remove the solvent, and a thin film is formed on the surface of the container. After preparing a liposome solution by adding a water-soluble solvent to the container on which the thin film is formed, it is transferred to a hermetic vial filled with the gas constituting the core and then mechanically mixed to produce MBs. After the generation of MBs, the residues dispersed in the solution are removed via centrifugation (Figure 3) [70,74,75].

### 3.3. Membrane Emulsification Method

MBs are formed at the pore outlet under a high-pressure state and are detached from the membrane by the shearing force applied by the solution containing the shell constituent material flowing in the continuous phase (Figure 4). The advantage of this method is that it produces MBs with a narrower size distribution than that of the aforementioned sonication and thin-film hydration methods. The size and size distribution depend on the membrane pore size, surface properties of the membrane, shear stress, transmembrane pressure, and substrate composing solution in the continuous phase [76,77].

### 3.4. Microfluidics Method

Although microfluidics has the advantage of generating narrow size distribution of MBs, it has restricted generation parameters and low productivity. The gas stream and the stream of the solution containing the shell component meet at the orifice, which is a feature of microfluidic devices. The MBs are then formed at a certain distance by a “pinch off” phenomenon (Figure 2). The generation of MBs is controlled by several parameters, such as the flow rate, viscosity ratios of gas and continuous phases, and the number of capillaries [78,79,80,81].

## 4. Utilizing Methods of MBs

MBs can be co-injected with various drugs or drug delivery platforms to increase the BBB permeability toward the drugs under ultrasound irradiation [82,83,84,85,86,87,88]. The co-injection method is advantageous, as it can be used together with the existing drug delivery systems and commercially available MBs to perform diverse treatments [89,90,91,92]. Moreover, MBs can be used as a drug delivery vehicle by loading a drug on a shell [63,86]. In this method, the MBs collapsed by ultrasound irradiation not only open the BBB but also release the loaded drugs, enabling stimuli-responsive drug release, which can increase the therapeutic effect. In this section, the MB-based platforms used for brain cancer therapy are summarized.

### 4.1. Co-Injection of MBs with Drugs

Co-injection of MBs and drugs induces a therapeutic effect against brain cancer by increasing BBB penetration. Commercially available MBs contrast agents, such as SonoVue^®^, Definity^®^, and Optison^®^, are used in the co-injection method that facilitates the delivery of cancer drugs, proteins, cells, and metal nanoparticles to the CNS [93,94,95,96]. MBs are co-injected with anticancer drugs to improve the therapeutic efficacy and diagnosis [91,92,97,98,99,100,101]. Kuo et al. confirmed the BBB opening effect by ultrasound and MBs on magnetic resonance imaging (MRI) [91]. Changes in the MRI intensity were observed after the co-injection of MRI contrast agents, gadopentetic acid (Gd-DTPA) and SonoVue^®^. The BBB opening effect by MBs and ultrasound enhanced the MR signal intensity by increasing the Gd-DTPA permeability. Additionally, the temozolomide (TMZ) concentration ratio in the cerebrospinal fluid (CSF) and plasma was observed to confirm the BBB-opening effect. The concentration ratio of TMZ in the CSF/plasma was 22.7% in the TMZ alone group, while that in the MB/ultrasound group increased to 33.6%. This result indicates that the BBB permeability of the drug increased owing to the BBB opening effect. Furthermore, Daniel et al. researched the effective treatment of brain cancer using SonoVue^®^ and paclitaxel (PTX) [102]. Although PTX is one of the anticancer drugs used in various cancer treatments, it has limited benefits in brain cancer due to its poor BBB permeability. The BBB-opening effect of MBs facilitates the penetration of PTX into the BBB and enhances its efficacy in brain cancer treatment. These results were observed in tumor-bearing mice. Tao et al. conducted an experiment to enhance drug delivery using Optison^®^ [103]. By observing the cavitation behavior of MBs, an experiment was conducted to prevent brain damage and deliver the drug effectively, and this was confirmed in an F98 rat glioma model. Hyungwon et al. studied the co-injection effect of SonoVue^®^ and drug-loaded ultrasound-sensitive liposomes [104]. An ultrasound-sensitive liposome is produced using the thin-film hydration method. Doxorubicin (Dox) is encapsulated in the liposome vesicles, which increase the drug biostability and release drugs in response to ultrasound. The released drug is delivered to the CNS by the BBB-opening effect of the MBs, facilitating chemotherapy for brain cancer.

The co-injection method of MBs is not limited to chemotherapy through the delivery of anticancer drugs but enables various therapies by allowing several substances to penetrate the CNS. Immune checkpoint inhibition (ICI) therapies, which target programmed cell death ligand 1 (PD-L1), programmed cell death-1, and cytotoxic T lymphocyte-associated protein-4, have been approved by the United States Food and Drug Administration for the treatment of various cancers. However, ICI therapies have limited efficacy in brain cancers [105,106]. Although the mechanisms underlying this failure remain unclear, the BBB is considered one of the obstacles in performing ICI therapy for brain cancer. Yan et al. investigated ICI therapy for brain cancer treatment using Definity^®^ and anti-PD-L1 (aPD-L1), which is an immune checkpoint inhibitor [107]. The effect of BBB opening by MBs and ultrasound increased the CNS delivery rate of aPD-L1, thus proving the possibility of applying ICI therapy in brain cancer cases. Moreover, it was confirmed that the injection method of MBs and control of the sonication pattern affect the delivery rate of aPD-L1 and the effects of ICI therapy. However, as the study was conducted on wild-type mice, additional in vivo experiments in brain cancer models are needed to verify this effect. Ryan et al. conducted experiments on BBB penetration of natural killer (NK)-92 cells for brain cancer therapy [108]. These cells are cytotoxic lymphocytes involved in the innate immune response against malignant cells. Decreased tumor size and increased survival rate in the rats were observed in the Definity^®^ and NK-92 cells treatment groups.

The BBB penetration by metal nanoparticles can be increased by the BBB-opening effect [109]. In the study by Dezhuan et al., MBs consisting of phospholipids, C_4_F_10_, and ^64^Cu-Au clusters were injected into the mice, and the BBB penetration effect was observed through positron emission tomography [110]. Pin et al. used carmustine-modified iron nanoparticles and SonoVue^®^ to facilitate MRI diagnosis and chemotherapy [111]. Carmustine is immobilized on the surface of iron nanoparticles by forming a covalent bond. The co-injection of the prepared nanoparticles and MBs facilitates the MRI monitoring of drug delivery and brain cancer therapy. This experiment was conducted using the B6 rat glioma model. Table 1 summarizes the types of MBs, constituent materials, drugs, cells, and animals used in the research that utilized the co-injection method.

### 4.2. MBs as Drug Delivery Vehicles

As mentioned in the aforementioned research, MBs can be used as a material for BBB opening as well as a drug delivery vehicle by conjugating various types of drugs with the MBs. The advantage of conjugating drugs with MBs is the prevention of the degradation of drugs that are unstable in the biological environment and increasing the clinical benefit even at reduced dosages. Furthermore, conjugated drug release is controlled by ultrasound-induced MBs destruction and facilitates targeted drug delivery [63,86]. Chien et al. produced drug-loaded MBs by encapsulating carmustine in the hydrophobic area of the MBs’ shells [64]. Carmustine was added during the preparation of the MBs consisting of C_3_F_8_, DPPC, and PEG-PL. The size of the prepared MBs was 1.11 μm, and their concentration was 19.78 ± 4.9 × 10^9^ MBs/mL. It was confirmed that a maximum of 1.67 mg of carmustine was loaded, and the drug encapsulation efficiency was 68.01 ± 4.35%. The MBs system increases the circulating half-life by increasing the biological stability of carmustine, which is easily hydrolyzed in the plasma, and by increasing the delivery rate into the CNS. Shih et al. observed that drug encapsulation and delivery efficiency depend on the hydrophobicity of the MBs’ shell, which was controlled by the length of the phospholipid alkyl chain [112]. The MBs were prepared using C_4_F_10_, DPPC, DSPC, and PEG-PL, and dextran was encapsulated in the MBs’ shell. In the delivery of low-molecular-weight (3 kDa) dextran, the hydrophobicity of the shell did not significantly affect the delivery; however, in the delivery of high-molecular-weight (40 kDa) dextran, the higher hydrophobicity of the shell, more effective is the dextran delivery. See et al. generated MBs using SF_6_ and a phospholipid with a thiol moiety and then conjugated it with a thiolated liposome containing Dox through a disulfide bond [70]. Following this, a peptide ligand targeting the interleukin 4 receptor, which is overexpressed in brain cancer, was post-modified on the MBs surface to improve the brain cancer-targeting effect. This system effectively targeted and treated U87 MG cancer cells. Boron neutron capture therapy (BNCT) has been receiving increasing attention for treating cancer. BNCT utilizes boronated agents (^10^B) to treat tumors, which, after undergoing irradiation with neutrons, yields ^7^Li and an alpha particle. Since the alpha particle has a short range, it causes irreversible damage to the DNA of the cancer cells without affecting the normal cells [113,114]. To accumulate a sufficient concentration of ^10^B for brain cancer treatment, it must be manufactured in the form of nanoparticles to utilize the enhanced permeability and retention effect. However, the BBB interrupts nanoparticle delivery to the CNS; thus, the effective delivery of ^10^B in brain cancer remains an important challenge in BNCT. Ching et al. modified the boron-containing polyanion (polyethylene glycol-b-poly((*closo*-dodecaboranyl)thiomethylstyrene) [PEG-b-PMBSH]) nanoparticles with MBs, which consists of C_3_F_8_, DPPC, and 1,2-dipalmitoyl-3-trimethylammonium-propane (DPTAP), via electrostatic interaction for BNCT [58]. This MBs-based platform can effectively deliver the PEG-b-PMBSH nanoparticles to the CNS under ultrasound irradiation.

In addition to hydrophobic cancer drugs, metal nanoparticles are encapsulated in the MBs to utilize the unique properties of the nanoparticles for brain cancer therapy. Ching et al. researched a system that facilitates chemotherapy and MRI diagnosis by modifying Dox and superparamagnetic iron oxide (SPIO) in MBs composed of DSPC, DSPG, PEG-PL, and C_3_F_8_ [63,115]. SPIO and Dox are conjugated by the reaction between the amine group functionalized on the SPIO surface and the carbonyl group of Dox, which are then encapsulated in the MBs’ shells [115]. In another study, Dox and SPIO were encapsulated in the MBs’ shells without further modification [63]. The MRI signal intensities and brain cancer therapy effects of the systems prepared using the two methods were confirmed in a C6 rat glioma model.

Biological materials with large structures, such as DNA and RNA, can penetrate the BBB by loading onto the MBs. Chang et al. prepared positively charged MBs using DPTAP to load negatively charged DNA onto the surface of the MBs through electrostatic interactions [116]. Additionally, vascular endothelial growth factor receptor 2 antibodies were modified onto the surface of the MBs through an avidin-biotin interaction to increase the gene therapy effect and enhance the cancer-targeting effect. The low BBB permeability of RNA is the main hurdle for utilizing RNA interference therapy against brain cancer. Guanjian et al. prepared the MBs and liposomes conjugated system using the host-guest interaction of avidin-biotin [67]. Before conjugation, short hairpin RNA, which decreases the transcription of the *BIRC5* gene, was loaded onto the liposomes. The effect of RNA interference therapy in brain cancer was confirmed in a C6 rat glioma model.

The use of MBs as a drug vehicle facilitates ultrasound-responsive drug release and increases BBB permeability through the opening effect of the MBs. Table 2 summarizes the types of MBs, manufacturing methods, constituent materials, and loaded drugs used for drug delivery.

## 5. Conclusions

In the past decade, significant progress has been made in improving the transport of therapeutics across the BBB through reversible opening via MB–ultrasound. This review describes the structure, composition, and preparation methods of the MBs. Hexafluoride (SF_6_) or perfluorocarbons, which have low solubility in an aqueous solution, were used as the MBs core to increase the BBB permeation. Various materials can be used as shells to reduce the high surface tension of MBs, but systems used for BBB permeation generally utilize the phospholipid derivatives and introduce PEG moiety to increase biocompatibility and stability. Methods for preparing MBs include sonication, thin-film hydration, membrane emulsification, and microfluidics methods. In the investigated research, sonication and thin-film hydration method were used to generate the MBs. Specifically, the information on drugs used together with MBs, which have recently been used for brain cancer therapy, has been summarized. MBs can be utilized by co-injection with a drug or by directly loading the drug onto them. Utilization of the MB-based systems and ultrasound facilitates the delivery of various medicines, such as anticancer drugs, proteins, liposomes, gene complexes, metal nanoparticles, and cells, to the tumor area in the CNS. Although this technology is in its initial investigation stage and must overcome many challenges before it can be used clinically, it has the potential to improve the therapeutic effect of various treatments that have not been attempted due to the BBB hindrance in the treatment of brain tumors.

## Data Availability

Not applicable.

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
