# Peer review of "Microbubble Delivery Platform for Ultrasound-Mediated Therapy in Brain Cancers"

_pharmaceutics, 2023, doi:10.3390/pharmaceutics15020698_

Round 1

Reviewer 1 Report

The manuscript is a review article about the use of microbubbles in opening the BBB under ultrasound.

In this review, the authors describe (i) the composition of microbbubles used in BBB opening, (ii) how these microbubbles are produced and (iii) how they are used in preclinical therapy (in combination with free drugs or as drug carriers).

It is a nice work where I have only a few critics:

line 38: I would also mention the presence of efflux pumps that are also efficient against drug.

lines 48-52: The two sentences suggest that some clinical trials involving BBB opening were performed without injected microbubbles. But as far as I know all clinical trials used i.v. injected microbubbles to open the BBB as given by the reference 24-26. If I am wrong, then the authors should provide references to these clinical trials. In addition, without the presence of the injected microbubbles, I do not see how stable cavitation can be achieved without inducing inertial cavitation as there is no control on the bubble size.

line 263, this should be 4.2 instead of 4.1

Author Response

Thank you for the valuable comment. We addressed the issues raised. We have clearly outlined how we addressed at below page.

Comments to the Author
The manuscript is a review article about the use of microbubbles in opening the BBB under ultrasound.

In this review, the authors describe (i) the composition of microbbubles used in BBB opening, (ii) how these microbubbles are produced and (iii) how they are used in preclinical therapy (in combination with free drugs or as drug carriers).

 It is a nice work where I have only a few critics:

Comment:

1) line 38: I would also mention the presence of efflux pumps that are also efficient against drug.

Answer: Manuscript is corrected as follows.

“However, barrier properties of the BBB, and efflux transporters P-glycoproteins and breast cancer resistant protein that are highly expressed in BBB are the primary cause hindering the delivery of drugs to the CNS for cancer treatment [12, 18].” (Page 1, line 37-38)

2) lines 48-52: The two sentences suggest that some clinical trials involving BBB opening were performed without injected microbubbles. But as far as I know all clinical trials used i.v. injected microbubbles to open the BBB as given by the reference 24-26. If I am wrong, then the authors should provide references to these clinical trials. In addition, without the presence of the injected microbubbles, I do not see how stable cavitation can be achieved without inducing inertial cavitation as there is no control on the bubble size.

Answer: Thank you for the valuable comment. Sentence is modified  as follows.

“Particularly, various clinical trials have been conducted to treat brain diseases effectively by promoting drug delivery to the CNS via control of the BBB through the stable and inertial cavitation phenomenon of microbubbles (MBs) that occur when focused ultrasound is applied to the cerebral blood vessels [24-26].” (Page 2, line 49-54)

3) line 263, this should be 4.2 instead of 4.1

Answer: The subsection number was corrected as informed by reviewer.

4.2. MBs as drug delivery vehicles” (Page 10, line 276)

Reviewer 2 Report

Myoung-Hwan Park et al. submitted a review manuscript “Microbubble delivery platform for ultrasound-mediated therapy in brain cancers”. The review is very well written, and provides an exhaustive overview of the MB-based delivery platforms in brain cancer therapy, which is accessible and logical. It summarizes the structure, composition, fabrication method and utilization of MBs. Especially, microbubble delivery platform with ultrasound can open the BBB and increase the therapeutic effect for brain cancer. 

There are some problems, which must be solved before it is considered for publication. Therefore, we recommend a major revision of this review. If the following problems are well-addressed, the reviewer believes that the essential contribution of this paper are important for brain cancers therapy.

1. Relevant research background ultrasound needs to be supplemented in INTRODUCTION.

2. In page 4, line 124, the subtitle “2.2” maybe should be “2.3”, and the same mistake in page 9, line 263, the subtitle “4.1” maybe should be “4.2”.

3. In page 4, the part “Sonication method” is non-exhaustive, and you should introduce it in a clear and straightforward manner.

4. In page 6, Scheme 4, the words are a bit blurry. Please consider replacing it with clearer one.

5. CONCLUSIONS needs more in it, as it's more of an repetition. The authors are suggested to highlight important findings and include prospect and bottlenecks of this work.

Author Response

Thank you for the valuable comment. We addressed the issues raised. We have clearly outlined how we addressed at below page.

Reviewer: 2

Comments to the Author
Myoung-Hwan Park et al. submitted a review manuscript “Microbubble delivery platform for ultrasound-mediated therapy in brain cancers”. The review is very well written, and provides an exhaustive overview of the MB-based delivery platforms in brain cancer therapy, which is accessible and logical. It summarizes the structure, composition, fabrication method and utilization of MBs. Especially, microbubble delivery platform with ultrasound can open the BBB and increase the therapeutic effect for brain cancer.

There are some problems, which must be solved before it is considered for publication. Therefore, we recommend a major revision of this review. If the following problems are well-addressed, the reviewer believes that the essential contribution of this paper are important for brain cancers therapy.

Comment:

1) Relevant research background ultrasound needs to be supplemented in INTRODUCTION.

Answer: Manuscript is corrected as follows.

“In the 1930s, Dr. Karl Dussik first published a paper on the use of ultrasound to visualize cerebral ventricles in the brain by measuring reflections of ultrasound through the head, and in the late 1960s, Dr. Claude Joyner first noted the development of MBs as a contrast medium, and the clinical use of MBs started after Gramiak and Shah acknowledged this subject [33-35].” (Page 2, line 57-61)

“In addition, since the required acoustic energy and response differ depending on the size and concentration of microbubbles, the physical parameters of ultrasound, such as period per pulse, pulse amplitude, pulse repetition frequency, and exposure length, must be ad-justed to determine the optimal ultrasound and microbubble combination [44,45].” (Page 2, line 68-72)

2) In page 4, line 124, the subtitle “2.2” maybe should be “2.3”, and the same mistake in page 9, line 263, the subtitle “4.1” maybe should be “4.2”.

Answer: Thank you for your valuable comment. The subsection number was corrected as informed by reviewer.

“2.3. Multi functionalization of MBs” (Page 4, line 131)

4.2. MBs as drug delivery vehicles” (Page 10, line 276)

3) In page 4, the part “Sonication method” is non-exhaustive, and you should introduce it in a clear and straightforward manner.

Answer: The sonication procedure for the generation of MBs was added in the manuscript as follows.

“The material used as the outer shell and dissolved in the aqueous solution, and sonication is applied. Gas is continuously supplied to the solution for MBs generation. After generating the MBs using sonication, centrifugation and/or filtration are required to remove the residual substances [34, 71-73].” (Page 7, line 187)

4) In page 6, Scheme 4, the words are a bit blurry. Please consider replacing it with clearer one.

 Answer: Scheme 4 was modification as follows.

Scheme 4. Illustration of the thin-membrane emulsification method (SPG, Shirasu porous glass) Reproduced with permission from [77] (Copyright © 2019, Elsevier B.V.)

5) CONCLUSIONS needs more in it, as it's more of an repetition. The authors are suggested to highlight important findings and include prospect and bottlenecks of this work.

“Hexafluoride (SF6) or perfluorocarbons, which have low solubility in an aqueous solution, were used as the MBs core to increase the BBB permeation. Various materials can be used as shells to reduce the high surface tension of MBs, but systems used for BBB permeation generally utilize the phospholipid derivatives and introduce PEG moiety to increase biocompatibility and stability. Methods for preparing MBs include sonication, thin-film hydration, membrane emulsification, and microfluidics methods. In the investigated research, sonication and thin-film hydration method were used to generate the MBs.” (Page 12, line 349-356)

Reviewer 3 Report

Applications of microbubbles in medicine is increasingly becoming noticed and it is attracting wide range of investigators. This review highlights works which uses microbubbles in drug delivery for brain treatments and facilitate drug passages through the blood brain barriers. 

I only recommend editing the manuscript especially language before publication.

As an example the sentence in line 15 of the abstract needs to be re-written. 

Also some comments in the attached document needs to be adhered to

Author Response

Thank you for the valuable comment. We addressed the issues raised. We have clearly outlined how we addressed at below page.

Reviewer: 3

Comments to the Author
Applications of microbubbles in medicine is increasingly becoming noticed and it is attracting wide range of investigators. This review highlights works which uses microbubbles in drug delivery for brain treatments and facilitate drug passages through the blood brain barriers.

Also some comments in the attached document needs to be adhered to

Comment:

1) I only recommend editing the manuscript especially language before publication. As an example, the sentence in line 15 of the abstract needs to be re-written.

Answer: This manuscript was edited through an English editing service before submission. The certification was attached to the final page. Line 15 of the abstract has been corrected to be more precise and concise.

“To overcome this challenge, microbubbles (MBs)-based drug delivery systems have garnered a lot of interest in recent years.” (Page 1, line 15)

2) what do you mean by "simple" air? do you mean at ospheric air?

Answer: Simple air was modified to atmospheric air as follows to convey the correct meaning.

“The first-generation MBs had low stability in solution because atmospheric air constituted the core, and they lacked a stabilizing shell.” (Page 3, line 90-91)

“Therefore, in brain cancer therapy, sulfur hexafluoride (SF6) or perfluorocarbons, which have higher molecular weights and lower blood solubility than those of atmospheric air, are used as the core in MBs.” (Page 3, line 94-96)

2) Complete this sentence. By loading where?

Answer: Sentence was modified as follows.

“Moreover, MBs can be used as a drug delivery vehicle by loading a drug on a shell [63, 86].” (Page 8, line 207)
